# Control Batch Size and Learning Rate to Generalize Well: Theoretical and Empirical Evidence

**Fengxiang He    Tongliang Liu    Dacheng Tao**
UBTECH Sydney AI Centre, School of Computer Science, Faculty of Engineering
The University of Sydney, Darlington, NSW 2008, Australia
`{fengxiang.he, tongliang.liu, dacheng.tao}@sydney.edu.au`

## Abstract

Deep neural networks have received dramatic success based on the optimization method of stochastic gradient descent (SGD). However, it is still not clear how to tune hyper-parameters, especially batch size and learning rate, to ensure good generalization. This paper reports both theoretical and empirical evidence of a training strategy that we should control the ratio of batch size to learning rate not too large to achieve a good generalization ability. Specifically, we prove a PAC-Bayes generalization bound for neural networks trained by SGD, which has a positive correlation with the ratio of batch size to learning rate. This correlation builds the theoretical foundation of the training strategy. Furthermore, we conduct a large-scale experiment to verify the correlation and training strategy. We trained 1,600 models based on architectures ResNet-110, and VGG-19 with datasets CIFAR-10 and CIFAR-100 while strictly control unrelated variables. Accuracies on the test sets are collected for the evaluation. Spearman's rank-order correlation coefficients and the corresponding $p$ values on 164 groups of the collected data demonstrate that the correlation is statistically significant, which fully supports the training strategy.

## 1   Introduction

The recent decade saw dramatic success of deep neural networks [9] based on the optimization method of stochastic gradient descent (SGD) [2, 32]. It is an interesting and important problem that how to tune the hyper-parameters of SGD to make neural networks generalize well. Some works have been addressing the strategies of tuning hyper-parameters [5, 10, 14, 15] and the generalization ability of SGD [4, 11, 19, 26, 27]. However, there still lacks solid evidence for the training strategies regarding the hyper-parameters for neural networks.

In this paper, we present both theoretical and empirical evidence for a training strategy for deep neural networks:

> When employing SGD to train deep neural networks, we should control the batch size not too large and learning rate not too small, in order to make the networks generalize well.

This strategy gives a guide to tune the hyper-parameters that helps neural networks achieve good test performance when the training error has been small. It is derived from the following property:

> The generalization ability of deep neural networks has a negative correlation with the ratio of batch size to learning rate.

As regards the theoretical evidence, we prove a novel PAC-Bayes [24, 25] upper bound for the generalization error of deep neural networks trained by SGD. The proposed generalization bound

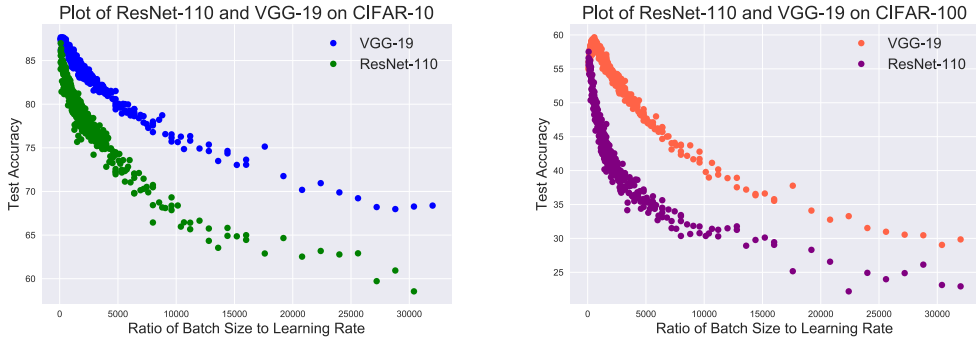

Figure 1: Scatter plots of accuracy on test set to ratio of batch size to learning rate. Each point represents a model. Totally 1,600 points are plotted.

has a positive correlation with the ratio of batch size to learning rate, which suggests a negative correlation between the generalization ability of neural networks and the ratio. This result builds the theoretical foundation of the training strategy.

From the empirical aspect, we conduct extensive systematic experiments while strictly control unrelated variables to investigate the influence of batch size and learning rate on the generalization ability. Specifically, we trained 1,600 neural networks based on two popular architectures, ResNet-110 [12, 13] and VGG-19 [28], on two standard datasets, CIFAR-10 and CIFAR-100 [16]. The accuracies on the test set of all the networks are collected for analysis. Since the training error is almost the same across all the networks (it is almost 0), the test accuracy is an informative index to express the generalization ability. Evaluation is then performed to 164 groups of the collected data. The Spearman's rank-order correlation coefficients and the corresponding $p$ values [31] demonstrate that the correlation is statistically significant (the probability that the correlation is wrong is smaller than 0.005), which fully supports the training strategy.

The rest of this paper is organized as follows. Section 2 recalls the preliminaries of generalization and SGD. Sections 3 and 4 respectively present the theoretical and empirical evidence of the training strategy. Section 5 reviews the related works. Section 6 concludes the paper. Appendix A presents additional background and preliminaries. Appendix B provides the proofs omitted from the main text.

## 2   Preliminaries

**Generalization bound for stochastic algorithms.** Formally, machine learning algorithms are designed to select the hypothesis function $F_\theta$ with the lowest expected risk $\mathcal{R}$ under the loss function $l$ from a hypothesis class $\{F_\theta | \theta \in \Theta \subset \mathbb{R}^d\}$, where $\theta$ is the parameter of the hypothesis and $d$ is the dimension of the parameter $\theta$. For many stochastic algorithms, such as SGD, we usually use a distribution to express the output parameter. Suppose the parameter follows a distribution $Q$, the expected risks respectively in terms of $\theta$ and $Q$ are defined as:

$$\mathcal{R}(\theta) = \mathbb{E}_{(X,Y)\sim\mathcal{D}} l(F_\theta(X), Y), \tag{1}$$

$$\mathcal{R}(Q) = \mathbb{E}_{\theta\sim Q}\mathbb{E}_{(X,Y)\sim\mathcal{D}} l(F_\theta(X), Y). \tag{2}$$

However, the expected risk $\mathcal{R}$ is not available from the data, since we do not know the formulation of latent distribution $\mathcal{D}$ of data. Practically, we use the empirical risk $\hat{\mathcal{R}}$ to estimate the expected risk $\mathcal{R}$, which is defined as:

$$\hat{\mathcal{R}}(\theta) = \frac{1}{|T|}\sum_{i=1}^{|T|} l(F_\theta(X_i), Y_i), \tag{3}$$

$$\hat{\mathcal{R}}(Q) = \mathbb{E}_{\theta\sim Q}\left[\frac{1}{|T|}\sum_{i=1}^{|T|} l(F_\theta(X_i), Y_i)\right], \tag{4}$$

where all $(X_i, Y_i)$ constitute the training sample $T$.

Equivalently, the empirical risk $\hat{\mathcal{R}}$ is the error of the algorithm on training data, while the expected risk $\mathcal{R}$ is the expectation of the error on test data or unseen data. Therefore, the difference between them is an informative index to express the generalization ability of the algorithm, which is called generalization error. The upper bound of the generalization error (usually called generalization bound) expresses how large the generalization error is possible to be. Therefore, generalization bound is also an important index to show the generalization ability of an algorithm.

**Stochastic gradient descent.** To optimize the expected risk (eq. 1), a natural tool is the gradient descent (GD). Specifically, the gradient of eq. (1) in terms of the parameter $\theta$ and the corresponding update equation are defined as follows,

$$g(\theta(t)) \triangleq \nabla_{\theta(t)} \mathcal{R}(\theta(t)) = \nabla_{\theta(t)} \mathbb{E}_{(X,Y)} l(F_{\theta(t)}(X), Y), \tag{5}$$

$$\theta(t+1) = \theta(t) - \eta g(\theta(t)), \tag{6}$$

where $\theta(t)$ is the parameter at the interation $t$ and $\eta > 0$ is the learning rate.

Stochastic gradient descent (SGD) estimates the gradient from mini batches of the training sample set to estimate the gradient $g(\theta)$. Let $S$ be the indices of a mini batch, in which all indices are independently and identically (i.i.d.) drawn from $\{1, 2, \ldots, N\}$, where $N$ is the training sample size. Then similar to the gradient, the iteration of SGD on the mini-batch $S$ is defined as follows,

$$\hat{g}_S(\theta(t)) = \nabla_{\theta(t)} \hat{\mathcal{R}}(\theta(t)) = \frac{1}{|S|} \sum_{n \in S} \nabla_{\theta(t)} l(F_{\theta(t)}(X_n), Y_n), \tag{7}$$

$$\theta(t+1) = \theta(t) - \eta \hat{g}(\theta(t)), \tag{8}$$

where $\hat{\mathcal{R}}(\theta) = \frac{1}{|S|} \sum_{n \in S} l(F_\theta(X_n), Y_n)$ is the empirical risk on the mini batch and $|S|$ is the cardinality of the set $S$. For brevity, we rewrite $l(F_\theta(X_n), Y_n) = l_n(\theta)$ in the rest of this paper.

Also, suppose that in step $i$, the distribution of parameter is $Q_i$, the initial distribution is $Q_0$, and the convergent distribution is $Q$. Then SGD is used to find $Q$ from $Q_0$ through a series of $Q_i$.

# 3 Theoretical Evidence

In this section, we explore and develop the theoretical foundations for the training strategy. The main ingredient is a PAC-Bayes generalization bound of deep neural networks based on the optimization method SGD. The generalization bound has a positive correlation with the ratio of batch size to learning rate. This correlation suggests the presented training strategy.

## 3.1 A Generalization Bound for SGD

Apparently, both $l_n(\theta)$ and $\hat{\mathcal{R}}(\theta)$ are un-biased estimations of the expected risk $\mathcal{R}(\theta)$, while $\nabla_\theta l_n(\theta)$ and $\hat{g}_S(\theta)$ are both un-biased estimations of the gradient $g(\theta) = \nabla_\theta \mathcal{R}(\theta)$:

$$\mathbb{E}\left[l_n(\theta)\right] = \mathbb{E}\left[\hat{\mathcal{R}}(\theta)\right] = \mathcal{R}(\theta), \tag{9}$$

$$\mathbb{E}\left[\nabla_\theta l_n(\theta)\right] = \mathbb{E}\left[\hat{g}_S(\theta)\right] = g(\theta) = \nabla_\theta \mathcal{R}(\theta), \tag{10}$$

where the expectations are in terms of the corresponding examples $(X, Y)$.

An assumption (see, e.g., [7, 23]) for the estimations is that all the gradients $\{\nabla_\theta l_n(\theta)\}$ calculated from individual data points are i.i.d. drawn from a Gaussian distribution centred at $g(\theta) = \nabla_\theta R(\theta)$:

$$\nabla_\theta l_n(\theta) \sim \mathcal{N}(g(\theta), C). \tag{11}$$

where $C$ is the covariance matrix and is a constant matrix for all $\theta$. As covariance matrices are (semi) positive-definite, for brevity, we suppose that $C$ can be factorized as $C = BB^\top$. This assumption can be justified by the central limit theorem when the sample size $N$ is large enough compared with the batch size $S$. Considering deep learning is usually utilized to process large-scale data, this assumption approximately holds in the real-life cases.

Therefore, the stochastic gradient is also drawn from a Gaussian distribution centred at $g(\theta)$:

$$\hat{g}_S(\theta) = \frac{1}{|S|}\sum_{n \in S}\nabla_\theta l_n(\theta) \sim \mathcal{N}\left(g(\theta), \frac{1}{|S|}C\right). \tag{12}$$

SGD uses the stochastic gradient $\hat{g}_S(\theta)$ to iteratively update the parameter $\theta$ in order to minimize the function $\mathcal{R}(\theta)$:

$$\Delta\theta(t) = \theta(t+1) - \theta(t) = -\eta\hat{g}_S(\theta(t)) = -\eta g(\theta) + \frac{\eta}{\sqrt{|S|}}B\Delta W,\ \Delta W \sim \mathcal{N}(0, I). \tag{13}$$

In this paper, we only consider the case that the batch size $|S|$ and learning rate $\eta$ are constant. Eq. (13) expresses a stochastic process which is well-known as Ornstein-Uhlenbeck process [33].

Furthermore, we assume that the loss function in the local region around the minimum is convex and 2-order differentiable:

$$\mathcal{R}(\theta) = \frac{1}{2}\theta^\top A\theta, \tag{14}$$

where $A$ is the Hessian matrix around the minimum and is a (semi) positive-definite matrix. This assumption has been primarily demonstrated by empirical works (see [18, p. 1, Figures 1(a) and 1(b) and p. 6, Figures 4(a) and 4(b)]). Without loss of generality, we assume that the global minimum of the objective function $\mathcal{R}(\theta)$ is 0 and achieves at $\theta = 0$. General cases can be obtained by translation operations, which would not change the geometry of objective function and the corresponding generalization ability. From the results of Ornstein-Uhlenbeck process, eq. (13) has an analytic stationary distribution:

$$q(\theta) = M\exp\left\{-\frac{1}{2}\theta^\top\Sigma^{-1}\theta\right\}, \tag{15}$$

where $M$ is the normalizer [8].

Estimating SGD as a continuous-time stochastic process dates back to works by [17, 21]. For a detailed justification, please refer to a recent work [see 23, pp. 6-8, Section 3.2].

We then obtain a generalization bound for SGD as follows.

**Theorem 1.** *For any positive real $\delta \in (0, 1)$, with probability at least $1 - \delta$ over a training sample set of size $N$, we have the following inequality for the distribution $Q$ of the output hypothesis function of SGD:*

$$R(Q) \le \hat{R}(Q) + \sqrt{\frac{\frac{\eta}{|S|}tr(CA^{-1}) - 2\log(\det(\Sigma)) - 2d + 4\log\left(\frac{1}{\delta}\right) + 4\log N + 8}{8N - 4}}, \tag{16}$$

*and*

$$\Sigma A + A\Sigma = \frac{\eta}{|S|}C, \tag{17}$$

*where $A$ is the Hessian matrix of the loss function around the local minimum, $B$ is from the covariance matrix of the gradients calculated by single sample points, and $d$ is the dimension of the parameter $\theta$ (network size).*

The proof for this generalization bound has two parts: (1) utilize results from stochastic differential equation (SDE) to find the stationary solution of the latent Ornstein-Uhlenbeck process (eq. 13) which expresses the iterative update of SGD; and (2) adapt the PAC-Bayes framework to obtain the generalization bound based on the stationary distribution. A detailed proof is omitted here and is given in Appendix B.1.

## 3.2 A Special Case of the Generalization Bound

In this subsection, we study a special case with two more assumptions for further understating the influence of the gradient fluctuation on our proposed generalization bound.

**Assumption 1.** *The matrices $A$ and $\Sigma$ are symmetric.*

Assumption 1 can be translated as that both the local geometry around the global minima and the stationary distribution are homogeneous to every dimension of the parameter space. Similar assumptions are also used by a recent work [14]. This assumption indicates that the product $\Sigma A$ of matrices $A$ and $\Sigma$ is also symmetric.

Based on Assumptions 1, we can further get the following theorem.

**Theorem 2.** *When Assumptions 1 holds, under all the conditions of Theorem 1, the stationary distribution of SGD has the following generalization bound,*

$$
\begin{aligned}
R(Q) \leq &\hat{R}(Q) \\
&+ \sqrt{\frac{\frac{\eta}{2|S|}tr(CA^{-1}) + d\log\left(\frac{2|S|}{\eta}\right) - \log(\det(CA^{-1})) - d + 2\log\left(\frac{1}{\delta}\right) + 2\log N + 4}{4N - 2}},
\end{aligned}
\tag{18}
$$

A detailed proof is omitted here and is given in Appendix B.2 in the supplementary materials.

Intuitively, our generalization bound links the generalization ability of the deep neural networks trained by SGD with three factors:

**Local geometry around minimum.** The determinant of the Hessian matrix $A$ expresses the local geometry of the objective function around the local minimum. Specifically, the magnitude of $\det(A)$ expresses the sharpness of the local minima. Many works suggest that sharp local minima relate to poor generalization ability [15, 10].

**Gradient fluctuation.** The covariance matrix $C$ (or equivalently the matrix $B$) expresses the fluctuation of the estimation to the gradient from individual data points which is the source of gradient noise. A recent intuition for the advantage of SGD is that it introduces noise into the gradient, so that it can jump out of bad local minima.

**Hyper-parameters.** Batch size $|S|$ and learning rate $\eta$, adjust the fluctuation of gradient. Specifically, under the following assumption, our generalization bound has a positive correlation with the ratio of batch size to learning rate.

**Assumption 2.** *The network size is large enough:*

$$
d > \frac{tr(CA^{-1})\eta}{2|S|},
\tag{19}
$$

*where $d$ is the number of the parameters, $C$ expresses the magnitude of individual gradient noise, $A$ is the Hessian matrix around the global minima, $\eta$ is the leaning rate, and $|S|$ is the batch size.*

This assumption can be justified that the network sizes of neural networks are usually extremely large. This property is also called *overparametrization* [6, 3, 1]. We can obtain the following corollary by combining Theorem 2 and Assumption 2.

**Corollary 1.** *When all conditions of Theorem 2 and Assumption 2 hold, the generalization bound of the network has a positive correlation with the ratio of batch size to learning rate.*

The proof is omitted from the main text and given in Appendix B.3

It reveals the negative correlation between the generalization ability and the ratio. This property further derives the training strategy that we should control the ratio not too large to achieve a good generalization when training deep neural networks using SGD.

## 4 Empirical Evidence

To evaluate the training strategy from the empirical aspect, we conduct extensive systematic experiments to investigate the influence of the batch size and learning rate on the generalization ability of deep neural networks trained by SGD. To deliver rigorous results, our experiments strictly control all unrelated variables. The empirical results show that there is a statistically significant negative correlation between the generalization ability of the networks and the ratio of the batch size to the learning rate, which builds a solid empirical foundation for the training strategy.

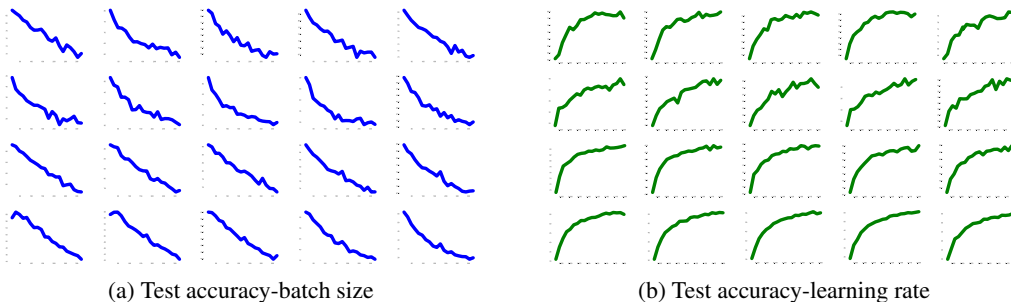

(a) Test accuracy-batch size           (b) Test accuracy-learning rate

Figure 2: Curves of test accuracy to batch size and learning rate. The four rows are respectively for (1) ResNet-110 trained on CIFAR-10, (2) ResNet-110 trained on CIFAR-100, (3) VGG-19 trained on CIFAR-10, and (4) VGG-19 trained on CIFAR-10. Each curve is based on 20 networks.

## 4.1 Implementation Details

To guarantee that the empirical results generally apply to any case, our experiments are conducted based on two popular architectures, ResNet-110 [12, 13] and VGG-19 [28], on two standard datasets, CIFAR-10 and CIFAR-100 [16], which can be downloaded from https://www.cs.toronto.edu/ kriz/cifar.html. The separations of the training sets and the test sets are the same as the official version.

We trained 1,600 models with 20 batch sizes, $S_{BS} = \{16, 32, 48, 64, 80, 96, 112, 128, 144, 160, 176,$ $192, 208, 224, 240, 256, 272, 288, 304, 320\}$, and 20 learning rates, $S_{LR} = \{0.01, 0.02, 0.03, 0.04,$ $0.05, 0.06, 0.07, 0.08, 0.09, 0.10, 0.11, 0.12, 0.13, 0.14, 0.15, 0.16, 0.17, 0.18, 0.19, 0.20\}$. All training techniques of SGD, such as momentum, are disabled. Also, both batch size and learning rate are constant in our experiments. Every model with a specific pair of batch size and learning rate is trained for 200 epochs. The test accuracies of all 200 epochs are collected for analysis. We select the highest accuracy on the test set to express the generalization ability of each model, since the training error is almost the same across all models (they are all nearly 0).

The collected data is then utilized to investigate three correlations: (1) the correlation between the generalization ability of networks and the batch size, (2) the correlation between the generalization ability of networks and the learning rate, and (3) the correlation between the generalization ability of networks and the ratio of batch size to learning rate, where the first two are preparations for the final one. Specifically, we calculate the Spearman's rank-order correlation coefficients (SCCs) and the corresponding $p$ value of 164 groups of the collected data to investigate the statistically significance of the correlations. Almost all results demonstrate the correlations are statistically significant ($p < 0.005$)[1]. The $p$ values of the correlation between the test accuracy and the ratio are all lower than $10^{-180}$ (see Table 3).

The architectures of our models are similar to a popular implementation of ResNet-110 and VGG-19[2]. Additionally, our experiments are conducted on a computing cluster with GPUs of NVIDIA® Tesla™ V100 16GB and CPUs of Intel® Xeon® Gold 6140 CPU @ 2.30GHz.

## 4.2 Empirical Results on the Correlation

**Correlation between generalization ability and batch size.** When the learning rate is fixed as an element of $S_{LR}$, we train ResNet-110 and VGG-19 on CIFAR10 and CIFAR100 with 20 batch sizes of $S_{BS}$. The plots of test accuracy to batch size are illustrated in Figure 2a. We list $1/4$ of all plots due to space limitation. The rest of the plots are in the supplementary materials. We then calculate the SCCs and the $p$ values as Table 1, where bold $p$ values refer to the statistically significant observations, while underlined ones refer to those not significant (as well as Table 2). The results clearly show that there is a statistically significant negative correlation between generalization ability and batch size.

Table 1: SCC and $p$ values of batch size to test accuracy for different learning rate (LR).

| LR | ResNet-110 on CIFAR-10 | | ResNet-110 on CIFAR-100 | | VGG-19 on CIFAR-10 | | VGG-19 on CIFAR-100 | |
|---|---|---|---|---|---|---|---|---|
| | SCC | $p$ | SCC | $p$ | SCC | $p$ | SCC | $p$ |
| 0.01 | $-0.96$ | $2.6 \times 10^{-11}$ | $-0.92$ | $5.6 \times 10^{-8}$ | $-0.98$ | $3.7 \times 10^{-14}$ | $-0.99$ | $7.1 \times 10^{-18}$ |
| 0.02 | $-0.96$ | $1.2 \times 10^{-11}$ | $-0.94$ | $1.5 \times 10^{-9}$ | $-0.99$ | $3.6 \times 10^{-17}$ | $-0.99$ | $7.1 \times 10^{-18}$ |
| 0.03 | $-0.96$ | $3.4 \times 10^{-11}$ | $-0.99$ | $1.5 \times 10^{-16}$ | $-0.99$ | $7.1 \times 10^{-18}$ | $-1.00$ | $1.9 \times 10^{-21}$ |
| 0.04 | $-0.98$ | $1.8 \times 10^{-14}$ | $-0.98$ | $7.1 \times 10^{-14}$ | $-0.99$ | $9.6 \times 10^{-19}$ | $-0.99$ | $3.6 \times 10^{-17}$ |
| 0.05 | $-0.98$ | $3.7 \times 10^{-14}$ | $-0.98$ | $1.3 \times 10^{-13}$ | $-0.99$ | $7.1 \times 10^{-18}$ | $-0.99$ | $1.4 \times 10^{-15}$ |
| 0.06 | $-0.96$ | $1.8 \times 10^{-11}$ | $-0.97$ | $6.7 \times 10^{-13}$ | $-1.00$ | $1.9 \times 10^{-21}$ | $-0.99$ | $1.4 \times 10^{-15}$ |
| 0.07 | $-0.98$ | $5.9 \times 10^{-15}$ | $-0.94$ | $5.0 \times 10^{-10}$ | $-0.98$ | $8.3 \times 10^{-15}$ | $-0.97$ | $1.7 \times 10^{-12}$ |
| 0.08 | $-0.97$ | $1.7 \times 10^{-12}$ | $-0.97$ | $1.7 \times 10^{-12}$ | $-0.98$ | $2.4 \times 10^{-13}$ | $-0.97$ | $1.7 \times 10^{-12}$ |
| 0.09 | $-0.97$ | $4.0 \times 10^{-13}$ | $-0.98$ | $3.7 \times 10^{-14}$ | $-0.98$ | $1.8 \times 10^{-14}$ | $-0.96$ | $1.2 \times 10^{-11}$ |
| 0.10 | $-0.97$ | $1.9 \times 10^{-12}$ | $-0.96$ | $8.7 \times 10^{-12}$ | $-0.98$ | $8.3 \times 10^{-15}$ | $-0.93$ | $2.2 \times 10^{-9}$ |
| 0.11 | $-0.97$ | $1.1 \times 10^{-12}$ | $-0.98$ | $1.3 \times 10^{-13}$ | $-0.99$ | $2.2 \times 10^{-16}$ | $-0.93$ | $2.7 \times 10^{-9}$ |
| 0.12 | $-0.97$ | $4.4 \times 10^{-12}$ | $-0.96$ | $2.5 \times 10^{-11}$ | $-0.98$ | $7.1 \times 10^{-13}$ | $-0.90$ | $7.0 \times 10^{-8}$ |
| 0.13 | $-0.94$ | $1.5 \times 10^{-9}$ | $-0.98$ | $1.3 \times 10^{-13}$ | $-0.97$ | $1.7 \times 10^{-12}$ | $-0.89$ | $1.2 \times 10^{-7}$ |
| 0.14 | $-0.97$ | $2.6 \times 10^{-12}$ | $-0.91$ | $3.1 \times 10^{-8}$ | $-0.97$ | $6.7 \times 10^{-13}$ | $-0.86$ | $1.1 \times 10^{-6}$ |
| 0.15 | $-0.96$ | $4.6 \times 10^{-11}$ | $-0.98$ | $1.3 \times 10^{-13}$ | $-0.95$ | $8.3 \times 10^{-11}$ | $-0.79$ | $3.1 \times 10^{-5}$ |
| 0.16 | $-0.95$ | $3.1 \times 10^{-10}$ | $-0.96$ | $8.7 \times 10^{-12}$ | $-0.95$ | $1.4 \times 10^{-10}$ | $-0.77$ | $6.1 \times 10^{-5}$ |
| 0.17 | $-0.95$ | $2.4 \times 10^{-10}$ | $-0.95$ | $2.6 \times 10^{-10}$ | $-0.91$ | $2.3 \times 10^{-8}$ | $-0.68$ | $1.3 \times 10^{-3}$ |
| 0.18 | $-0.97$ | $4.0 \times 10^{-12}$ | $-0.97$ | $1.1 \times 10^{-12}$ | $-0.93$ | $2.6 \times 10^{-9}$ | $-0.66$ | $2.8 \times 10^{-3}$ |
| 0.19 | $-0.94$ | $6.3 \times 10^{-10}$ | $-0.95$ | $8.3 \times 10^{-11}$ | $-0.90$ | $8.0 \times 10^{-8}$ | $-0.75$ | $3.4 \times 10^{-4}$ |
| 0.20 | $-0.91$ | $3.6 \times 10^{-8}$ | $-0.98$ | $1.3 \times 10^{-13}$ | $-0.95$ | $6.2 \times 10^{-11}$ | $-0.57$ | $\underline{1.4 \times 10^{-2}}$ |

**Correlation between generalization ability and learning rate.** When the batch size is fixed as an element of $S_{BS}$, we train ResNet-110 and VGG-19 on CIFAR10 and CIFAR100 respectively with 20 learning rates $S_{LR}$. The plot of the test accuracy to the learning rate is illustrated in Figure 2b, which include $1/4$ of all plots due to space limitation. The rest of the plots are in the supplementary materials. We then calculate the SCC and the $p$ values as Table 2 shows. The results clearly show that there is a statistically significant positive correlation between the learning rate and the generalization ability of SGD.

Table 2: SCC and $p$ values of learning rate to test accuracy for different batch size (BS).

| BS | ResNet-110 on CIFAR-10 | | ResNet-110 on CIFAR-100 | | VGG-19 on CIFAR-10 | | VGG-19 on CIFAR-100 | |
|---|---|---|---|---|---|---|---|---|
| | SCC | $p$ | SCC | $p$ | SCC | $p$ | SCC | $p$ |
| 16 | 0.60 | $\underline{5.3 \times 10^{-3}}$ | 0.84 | $3.2 \times 10^{-6}$ | 0.62 | $3.4 \times 10^{-3}$ | $-0.80$ | $2.6 \times 10^{-5}$ |
| 32 | 0.60 | $\underline{5.0 \times 10^{-3}}$ | 0.90 | $9.9 \times 10^{-8}$ | 0.78 | $4.9 \times 10^{-5}$ | $-0.14$ | $\underline{5.5 \times 10^{-1}}$ |
| 48 | 0.84 | $3.2 \times 10^{-6}$ | 0.89 | $1.8 \times 10^{-7}$ | 0.87 | $4.9 \times 10^{-7}$ | 0.37 | $\underline{1.1 \times 10^{-1}}$ |
| 64 | 0.67 | $1.2 \times 10^{-3}$ | 0.89 | $1.0 \times 10^{-7}$ | 0.91 | $2.0 \times 10^{-8}$ | 0.91 | $\underline{1.1 \times 10^{-6}}$ |
| 80 | 0.80 | $2.0 \times 10^{-5}$ | 0.99 | $4.8 \times 10^{-16}$ | 0.95 | $2.4 \times 10^{-10}$ | 0.87 | $4.5 \times 10^{-6}$ |
| 96 | 0.79 | $3.3 \times 10^{-5}$ | 0.89 | $2.4 \times 10^{-7}$ | 0.94 | $5.2 \times 10^{-9}$ | 0.94 | $1.5 \times 10^{-9}$ |
| 112 | 0.90 | $8.8 \times 10^{-8}$ | 0.91 | $2.7 \times 10^{-8}$ | 0.97 | $2.6 \times 10^{-12}$ | 0.95 | $1.2 \times 10^{-10}$ |
| 128 | 0.95 | $8.3 \times 10^{-11}$ | 0.92 | $1.1 \times 10^{-8}$ | 0.98 | $2.2 \times 10^{-14}$ | 0.99 | $4.8 \times 10^{-16}$ |
| 144 | 0.85 | $2.1 \times 10^{-6}$ | 0.98 | $7.7 \times 10^{-14}$ | 0.90 | $6.2 \times 10^{-8}$ | 0.98 | $3.5 \times 10^{-15}$ |
| 160 | 0.90 | $4.3 \times 10^{-8}$ | 0.94 | $5.0 \times 10^{-10}$ | 0.95 | $3.3 \times 10^{-10}$ | 0.99 | $7.1 \times 10^{-18}$ |
| 176 | 0.94 | $5.0 \times 10^{-10}$ | 0.99 | $3.6 \times 10^{-17}$ | 0.91 | $2.3 \times 10^{-8}$ | 0.98 | $1.8 \times 10^{-14}$ |
| 192 | 0.94 | $6.7 \times 10^{-10}$ | 0.94 | $5.0 \times 10^{-10}$ | 0.95 | $6.2 \times 10^{-11}$ | 0.97 | $2.6 \times 10^{-12}$ |
| 208 | 0.91 | $3.6 \times 10^{-8}$ | 0.97 | $6.7 \times 10^{-12}$ | 0.98 | $6.1 \times 10^{-14}$ | 0.99 | $2.5 \times 10^{-17}$ |
| 224 | 0.90 | $9.0 \times 10^{-8}$ | 0.98 | $3.7 \times 10^{-14}$ | 0.93 | $2.2 \times 10^{-9}$ | 0.98 | $1.3 \times 10^{-13}$ |
| 240 | 0.78 | $4.6 \times 10^{-5}$ | 0.95 | $2.4 \times 10^{-10}$ | 0.98 | $8.3 \times 10^{-15}$ | 0.99 | $9.6 \times 10^{-19}$ |
| 256 | 0.83 | $4.8 \times 10^{-6}$ | 0.94 | $5.0 \times 10^{-10}$ | 0.99 | $4.8 \times 10^{-16}$ | 0.97 | $5.4 \times 10^{-12}$ |
| 272 | 0.95 | $2.4 \times 10^{-10}$ | 0.96 | $2.5 \times 10^{-11}$ | 0.97 | $4.0 \times 10^{-13}$ | 0.99 | $1.5 \times 10^{-16}$ |
| 288 | 0.94 | $9.8 \times 10^{-10}$ | 0.92 | $1.5 \times 10^{-18}$ | 0.95 | $8.3 \times 10^{-11}$ | 0.99 | $1.5 \times 10^{-16}$ |
| 304 | 0.81 | $1.5 \times 10^{-5}$ | 0.97 | $4.0 \times 10^{-13}$ | 0.95 | $6.2 \times 10^{-11}$ | 1.00 | $3.7 \times 10^{-24}$ |
| 320 | 0.94 | $1.4 \times 10^{-9}$ | 0.95 | $8.3 \times 10^{-11}$ | 0.97 | $2.6 \times 10^{-12}$ | 1.00 | $7.2 \times 10^{-20}$ |

**Correlation between generalization ability and ratio of batch size to learning rate.** We plot the test accuracies of ResNet-110 and VGG-19 on CIFAR-10 and CIFAR-100 to the rate of batch size to learning rate in Figure 1. Totally over 1,600 points are plotted. Additionally, we perform Spearman's rank-order correlation test on all the accuracies of ResNet-110 and VGG-19 on CIFAR-10 and CIFAR-100. The SCC and $p$ values show that the correlation between the ratio and the generalization

ability is statistically significant as Table 3 demonstrate. Each test is performed on 400 models. The results strongly support the training strategy.

Table 3: SCC and $p$ values of the ratio of batch size to learning rate and test accuracy.

| ResNet-110 on CIFAR-10 | | ResNet-110 on CIFAR-100 | | VGG-19 on CIFAR-10 | | VGG-19 on CIFAR-100 | |
|---|---|---|---|---|---|---|---|
| SCC | $p$ | SCC | $p$ | SCC | $p$ | SCC | $p$ |
| $-0.97$ | $\mathbf{3.3 \times 10^{-235}}$ | $-0.98$ | $\mathbf{5.3 \times 10^{-293}}$ | $-0.98$ | $\mathbf{6.2 \times 10^{-291}}$ | $-0.94$ | $\mathbf{6.1 \times 10^{-180}}$ |

## 5   Related Work

Former experiments have been addressing the influence of batch size and learning rate on the generalization ability; however, the influence of hyper-parameters on the generalization is still under debate. [15] uses experiments to show that the large-batch training leads to sharp local minima which have the poor generalization of SGD, while small-batches lead to flat minima which make SGD generalize well. However, only two values of batch size are investigated. [10] proposes the Linear Scaling Rule to maintain the generalization ability of SGD: "When the minibatch size is multiplied by $k$, multiply the learning rate by $k$". [14] conducts experiments to show that the dynamics, geometry, and the generalization are dependent on the ratio $|S|/\eta$, but both the batch size and learning rate only have two values which considerably limits the generality of the results. Meanwhile, [5] finds that most current notions for sharpness/flatness are ill-defined. [30] proves that for linear neural networks, batch size has an optimal value when the learning rate is fixed. [29] suggests that increasing the batch size during the training of neural networks can receive the same result of decaying the learning rate when the rate of batch size to learning rate remains the same. This result is consistent with ours.

Some generalization bounds for algorithms trained by SGD are proposed, but they have not derived hyper-parameters tuning strategies. [26] studies the generalization of stochastic gradient Langevin dynamics (SGLD), and proposes an $\mathcal{O}\left(1/N\right)$ generalization bound and an $\mathcal{O}\left(1/\sqrt{N}\right)$ generalization bound respectively via the stability and the PAC-Bayesian theory. [27] studies the generalization of all iterative and noisy algorithms and gives a generalization error bound based on the mutual information between the input data and the output hypothesis. As exemplary cases, it gives generalization bounds for some special cases like SGLD. [4] proves a trade-off between convergence and stability for all iterative algorithms respectively under convex smooth setting and strong convex smooth setting. Considering the equivalence between stability and generalization, it also gives a trade-off between the convergence and the generalization. Under the same assumptions, [4] gives an $\mathcal{O}\left(1/N\right)$ generalization bound. [20] proves $\mathcal{O}(1/N)$ bounds for SGD when the loss function is Lipschitz continuous and smooth. [22] also gives a PAC-Bayes generalization bound for SGD based on the KL divergence of posterior $Q$ and the prior $P$ which is somewhat premature.

## 6   Conclusion

This work presents a training strategy for stochastic gradient descent (SGD) in order to achieve a good generalization ability: we should control the ratio of batch size to learning rate not too large while tuning the hyper-parameters. This strategy is based on a negative correlation between the generalization ability of networks with the ratio of batch size to learning rate, which is proved from both theoretical and empirical aspects in this paper. As the theoretical evidence, we prove a novel PAC-Bayes upper bound for the generalization error of algorithms trained by SGD. The bound has a positive correlation with the ratio of batch size to learning rate, which suggests a negative correlation between the generalization ability the ratio. For the empirical evidence, we trained 1,600 models based on ResNet-110 and VGG-19 on CIFAR-10 and CIFAR-110 and collected the accuracy on the test sets, while strictly control other variables. Spearman's order-rank correlation coefficients and the corresponding $p$ values are then calculated on 164 groups of the collected data. The results demonstrate that the correlation is statistically significant, which is in full agreement with the training strategy.

**Acknowledgments**

This work was supported in part by Australian Research Council Projects FL-170100117, DP180103424, and DE190101473.

## Footnotes

[1]The definition of "statistically significant" has various versions, such as $p < 0.05$ and $p < 0.01$. This paper uses a more rigorous one ($p < 0.005$).

[2]See Wei Yang, https://github.com/bearpaw/pytorch-classification, 2017.

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
