[Supplementary Material · 676-appendix.pdf]

# A  Background

## A.1   Convex and 2-Order Differentiable Assumptions

In Section 3.1, we use an assumption that the local minima are convex and 2-order differentiable to derive our generalization bound. This assumption has been primarily proved by empirical works (see fig. 3 which is originally presented in a recent work by [18, p. 1, figs. 1(a) and 1(b) and p. 6, figs. 4(a) and 4(b)]).

(a) ResNet-56 with skips      (b) DenseNet (112 layers) with skips

Figure 3: The risk surfaces with/without skips ([18]).

## A.2   The PAC-Bayesian Framework

The PAC-Bayesian framework dates back to works by [24, 25]. In the PAC-Bayesian view, the hypothesis function learned by a stochastic machine learning algorithm is drawn randomly (but still under several "laws") from the hypothesis class. The generalization capability of the algorithm has a negative correlation with the distance (usually measured by Kullback-Leibler (KL) divergence) between the distribution of the output hypothesis and the priori (usually a Gaussian distribution or uniform distribution). This result gives a trade-off between minimising the empirical risk and exploring further areas of the hypothesis space from the initial (priori).

Suppose the prior distribution over the parameter space $\Theta$ is $P$. Let $Q$ is the distribution on the parameter space $\Theta$ expressing the learned hypothesis function. We then define the expected risk with respect to the distribution $Q$ is as follows

$$\mathcal{R}(Q) = \mathbb{E}_{\theta \sim Q} \mathcal{R}(\theta).$$

Similarly, the empirical risk with respect to the distribution $Q$ is defined as

$$\hat{\mathcal{R}}(Q) = \mathbb{E}_{\theta \sim Q} \hat{\mathcal{R}}(\theta).$$

Then, a classic result uniformly bounding the expected risk $\mathcal{R}(Q)$ in terms of the empirical risk $\hat{\mathcal{R}}(Q)$ and the KL divergence $\mathcal{D}(Q||P)$ is as follows.

**Lemma 1** (see [24], Theorem 1). *For any positive real $\delta \in (0,1)$, with probability at least $1 - \delta$ over a sample of size $N$, we have the following inequality for all distributions $Q$:*

$$\mathcal{R}(Q) \leq \hat{\mathcal{R}}(Q) + \sqrt{\frac{\mathcal{D}(Q||P) + \log\frac{1}{\delta} + \log N + 2}{2N - 1}}, \tag{20}$$

*where $\mathcal{D}(Q||P)$ is the KL divergence between the distributions $Q$ and $P$ and is defined as,*

$$\mathcal{D}(Q||P) = \mathbb{E}_{\theta \sim Q} \left( \log \frac{Q(\theta)}{P(\theta)} \right). \tag{21}$$

# B  Proofs

To obtain the generalization bound, we follow an emerging and promising path of using a stochastic differential equation to model the iterative updates of SGD (see, e.g., [7, 23, 26]). It is indicated that

the update equation of SGD can be translated as Ornstein-Uhlenbeck process [33] under appropriate assumptions. Ornstein-Uhlenbeck process has an analytic stationary distribution, with which we can express the distribution of the weights of the output hypothesis function. By exploiting the stationary distribution, we further obtain a generalization bound via the stationary distribution by employing the PAC-Bayesian framework which presents a negative correlation between the generalization of a stochastic algorithm and the distance between the distribution of the output hypothesis and the prior distribution on the hypothesis space [24, 25].

To avoid technicalities, the measurability/integrability issues are ignored throughout this paper. Moreover, Fubini's theorem is assumed to be applicable for any integration with respect to multiple variables, that the order of integrations is exchangeable. Also, we assume the stable (stationary) solutions of all stochastic differential equations involved exit and are unique.

## B.1 Proof of Theorem 1

**Lemma 2** (cf. [23], pp. 27-18, Appendix B). *Under the 2-order differentiable assumption (eq. 14), the Ornstein-Uhlenbeck process (eq. 13)'s stationary distribution,*

$$q(\theta) = M \exp\left\{-\frac{1}{2}\theta^\top \Sigma^{-1} \theta\right\}, \tag{22}$$

*has the following property,*

$$A\Sigma + \Sigma A = \frac{\eta}{|S|} C. \tag{23}$$

This lemma is from [23]. Here, we recall the proof to make this paper complete.

*Proof.* Form a result in Ornstein-Uhlenbeck process [8], we know that the parameter $\theta$ has the following analytic solution,

$$\theta(t) = \theta(0)e^{-At} + \sqrt{\frac{\eta}{|S|}} \int_0^t e^{-A(t-t')} B\mathrm{d}W(t'), \tag{24}$$

where $W(t')$ is a white noise and follows $\mathcal{N}(0, I)$. From eq. (22), we know that

$$\Sigma = \mathbb{E}_{\theta \sim Q}\left[\theta\theta^\top\right]. \tag{25}$$

Therefore, we have the following equation,

$$
\begin{aligned}
A\Sigma + \Sigma A =& \frac{\eta}{|S|} \int_{-\infty}^t Ae^{-A(t-t_0)} Ce^{-A(t-t_0)} \mathrm{d}t' \\
& + \frac{\eta}{|S|} \int_{-\infty}^t e^{-A(t-t_0)} Ce^{-A(t-t_0)} \mathrm{d}t' A \\
=& \frac{\eta}{|S|} \int_{-\infty}^t \frac{\mathrm{d}}{\mathrm{d}t'} Ae^{-A(t-t_0)} Ce^{-A(t-t_0)} \\
=& \frac{\eta}{|S|} C.
\end{aligned}
\tag{26}
$$

The proof is completed. □

*Proof of Theorem 1.* In PAC-Bayesian framework (Lemma 1), an essential part is the KL divergence between the distribution of the learned hypothesis and the priori on the hypothesis space. The prior distribution can be interpreted as the distribution of the initial parameters, which are usually settled according to Gaussian distributions or uniform distributions.[3] Here, we use a standard Gaussian

distribution $\mathcal{N}(0, I)$ as the priori. Suppose the densities of the stationary distribution $Q$ and the prior distribution $P$ are respectively $p(\theta)$ and $q(\theta)$ in terms of the parameter $\theta$ as the following equations,

$$p(\theta) = \frac{1}{\sqrt{2\pi \det(I)}} \exp\left\{-\frac{1}{2}\theta^\top I \theta\right\}, \tag{27}$$

$$q(\theta) = \frac{1}{\sqrt{2\pi \det(\Sigma)}} \exp\left\{-\frac{1}{2}\theta^\top \Sigma^{-1} \theta\right\}, \tag{28}$$

where ep. (28) comes from eq. (22) by calculating the normalizer $M$.

Therefore,

$$
\begin{aligned}
&\log\left(\frac{q(\theta)}{p(\theta)}\right) \\
&= \log\left(\frac{\sqrt{2\pi \det(I)}}{\sqrt{2\pi \det(\Sigma)}} \exp\left\{\frac{1}{2}\theta^\top I\theta - \frac{1}{2}\theta^\top \Sigma^{-1}\theta\right\}\right) \\
&= \frac{1}{2}\log\left(\frac{1}{\det(\Sigma)}\right) + \frac{1}{2}\left(\theta^\top I\theta - \theta^\top \Sigma^{-1}\theta\right).
\end{aligned} \tag{29}
$$

Applying eq. (29) to eq. (21), we can calculate the KL divergence between the distributions $Q$ and $P$ (we assume $\Theta = \mathbb{R}^d$):

$$
\begin{aligned}
&\mathcal{D}(Q\|P) \\
&= \mathbb{E}_{\theta \sim Q}\left(\log\frac{Q(\theta)}{P(\theta)}\right) \\
&= \int_{\theta \in \Theta} \log\left(\frac{q(\theta)}{p(\theta)}\right) q(\theta)\mathrm{d}\theta \\
&= \int_{\theta \in \Theta} \left[\frac{1}{2}\log\left(\frac{1}{\det(\Sigma)}\right) + \frac{1}{2}\left(\theta^\top I\theta - \theta^\top \Sigma^{-1}\theta\right)\right] q(\theta)\mathrm{d}\theta \\
&= \frac{1}{2}\log\left(\frac{1}{\det(\Sigma)}\right) + \frac{1}{2}\int_{\theta \in \Theta} \theta^\top I\theta p(\theta)\mathrm{d}\theta - \frac{1}{2}\int_{\mathbb{R}^{|S|}} \theta^\top \Sigma^{-1}\theta q(\theta)\mathrm{d}\theta \\
&= \frac{1}{2}\log\left(\frac{1}{\det(\Sigma)}\right) + \frac{1}{2}\mathbb{E}_{\theta \sim \mathcal{N}(0,\Sigma)}\theta^\top I\theta - \frac{1}{2}\mathbb{E}_{\theta \sim \mathcal{N}(0,\Sigma)}\theta^\top \Sigma^{-1}\theta \\
&= \frac{1}{2}\log\left(\frac{1}{\det(\Sigma)}\right) + \frac{1}{2}\mathrm{tr}(\Sigma - I).
\end{aligned} \tag{30}
$$

From eq. (23), we have that

$$A\Sigma + \Sigma A = \frac{\eta}{|S|}C. \tag{31}$$

Therefore,

$$A\Sigma A^{-1} + \Sigma = \frac{\eta}{|S|}CA^{-1}. \tag{32}$$

After calculating the trace of the both sides, we have the following equation,

$$\mathrm{tr}\left(A\Sigma A^{-1} + \Sigma\right) = \mathrm{tr}\left(\frac{\eta}{|S|}CA^{-1}\right). \tag{33}$$

The left-hand side (LHS) is as follows,

$$
\begin{aligned}
\mathrm{LHS} &= \mathrm{tr}\left(A\Sigma A^{-1} + \Sigma\right) \\
&= \mathrm{tr}\left(A\Sigma A^{-1}\right) + \mathrm{tr}\left(\Sigma\right) \\
&= \mathrm{tr}\left(\Sigma A^{-1}A\right) + \mathrm{tr}\left(\Sigma\right) \\
&= \mathrm{tr}\left(\Sigma\right) + \mathrm{tr}\left(\Sigma\right) \\
&= 2\mathrm{tr}\left(\Sigma\right).
\end{aligned} \tag{34}
$$

Therefore,

$$\text{tr}\,(\Sigma) = \frac{1}{2}\text{tr}\left(\frac{\eta}{|S|}CA^{-1}\right) = \frac{1}{2}\frac{\eta}{|S|}\text{tr}\left(CA^{-1}\right). \tag{35}$$

At the same time, we can easily calculate that

$$\text{tr}(I) = d, \tag{36}$$

as $I \in \mathbb{R}^{d\times d}$, where $d$ is the dimension of the parameter $\theta$.

Insert eqs. (35) and (36) to eq. (30), we can get the following inequality,

$$\mathcal{D}(Q||P) \leq \frac{1}{4}\frac{\eta}{|S|}tr(CA^{-1}) - \frac{1}{2}\log(\det(\Sigma)) - \frac{1}{2}d. \tag{37}$$

Eq. (37) gives an upper bound for the distance (measured by KL divergence) between the stationary distribution of the output weights by SGD and the priori on the hypothesis space. Considering the monotonicity of the generalization bound in terms of the KL divergence, we can further obtain a PAC-Bayesian generalization bound for SGD by inserting the KL divergence bound (eq. 37) into the PAC-Bayesian framework (eq. (20) of Lemma 1).

The proof is completed. □

### B.2 Proof of Theorem 2

This subsection gives a proof for Theorem 2. Before proving it, we first present a Lemma.

**Lemma 3.** *When Assumptions 1, the KL divergence between the stationary distribution $Q$ of SGD and the prior distribution $P$ is satisfies the following inequality*

$$\mathcal{D}(Q||P) \leq \frac{\eta}{4|S|}tr(CA^{-1}) + \frac{1}{2}d\log\left(\frac{2|S|}{\eta}\right) - \frac{1}{2}\log(\det(CA^{-1})) - \frac{1}{2}d. \tag{38}$$

Lemma 3 gives an upper bound for the distance between the distribution of the output hypothesis by SGD and the prior distribution of the hypothesis space. It measures how far SGD can explore in the hypothesis space. Based on it, we can further get the following theorem that controls the generalization error of the special case under Assumptions 1.

*Proof of Lemma 3.* Apply Assumptions 1 to eq. (23), we can get the following equation.

$$\Sigma A + A\Sigma = \frac{\eta}{|S|}C$$

$$2\Sigma A = \frac{\eta}{|S|}C$$

$$\Sigma = \frac{\eta}{2|S|}CA^{-1}. \tag{39}$$

Therefore,

$$\det(\Sigma) = \det\left(\frac{\eta}{2|S|}CA^{-1}\right) = \left(\frac{\eta}{2|S|}\right)^d \det\left(CA^{-1}\right). \tag{40}$$

Thus,

$$\log\left(\det(\Sigma)\right) = \log\left[\left(\frac{\eta}{2|S|}\right)^d \det\left(CA^{-1}\right)\right]$$

$$= -d\log\left(\frac{2|S|}{\eta}\right) + \log\left[\det\left(CA^{-1}\right)\right]. \tag{41}$$

Applying eqs. (39) and (36) to eq. (30), we can get eq. (38).

The proof is completed. □

Then, we can directly obtain Theorem 2.

*Proof of Theorem 2.* Apply eq. (38) of Lemma 3 to eq. (20) of Lemma 1 of Lemma 1, we can directly get eq. (18).

The proof is completed. □

### B.3 Proof of Corollary 1

*Proof of Corollary 1.* We first define

$$I = \frac{\eta}{2|S|} tr(CA^{-1}) + d \log\left(\frac{2|S|}{\eta}\right) - \log(\det(CA^{-1})) - d + 2\log\left(\frac{1}{\delta}\right) + 2\log N + 4. \quad (42)$$

Then the generalization eq. (18) becomes,

$$R(Q) \leq \hat{R}(Q) + \sqrt{\frac{I}{4N-2}}, \quad (43)$$

We thus calculate the derivative of $I$ with respect to the ratio $\frac{|S|}{\eta}$ in order to check whether the generalization bound has a positive correlation withe the ratio. For the brevity, we define $k = \frac{|S|}{\eta}$.

$$\begin{aligned}
\frac{\partial I}{\partial k} &= \frac{\partial}{\partial k}\left[\frac{1}{2k} tr(CA^{-1}) + d\log(2k) - \log(\det(CA^{-1})) - d + 2\log\left(\frac{1}{\delta}\right) + 2\log N + 4\right] \\
&= \frac{\partial}{\partial k}\left[\frac{1}{2k} tr(CA^{-1}) + d\log(2k) - \log(\det(CA^{-1})) - d + 2\log\left(\frac{1}{\delta}\right) + 2\log N + 4\right] \\
&= -\frac{1}{2k^2} tr(CA^{-1}) + \frac{d}{k}
\end{aligned} \quad (44)$$

Therefore, when Assumption 2 holds, we have

$$d > \frac{tr(CA^{-1})\eta}{2|S|} = \frac{1}{2k} tr(CA^{-1}). \quad (45)$$

Thus,

$$\frac{\partial I}{\partial k} > 0. \quad (46)$$

So, $I$ and further the generalization bound has a positive correlation with the ratio of batch size to learning rate.

The proof is completed. □

## Footnotes

[3]Usually, when there is no confident prior knowledge of the latent model parameters, the priori should be set as distributions with no information, such as Gaussian distributions or uniform distributions. This setting comes from two considerations: (1) Once the algorithms based on the Bayesian statistics can converge, after long enough time and with big enough data, the algorithms can always converge to the stationary distributions. This is guaranteed by the assumption that the stationary solution of the latent stochastic differential equation exists and is unique; (2) Setting priori should be very careful, as we can not assume we have any knowledge of the target hypothesis function before we have started training the model.