[Reviews · NeurIPS 2019]

Reviewer 1



Theory-wise, the authors overlooked to discuss several prior works, some of which suggested opposite theories to theirs. For example: - "Don't Decay the Learning Rate, Increase the Batch Size", ICLR'18, seems to support a constant batch size/lr ratio empirically --- after rebuttal --- After reading the comments and the authors rebuttal, I am satisfied with the responses. The paper theoretically verifies that the ratio of batch size to learning rate is positively related to the generalization error. Specifically, it verifies some very recent empirical findings, e.g., Don’t decay the learning rate, increase the batch size, ICLR 2018, which empirically states that increasing the batch size and decaying the learning rate are quantitatively equivalent. I think the theoretical result is novel and timely and would interest many readers in the deep learning community. I value the theoretical contribution and thus would like increase my score and vote for accepting the submission. Experiment-wise, I have lots of reservations in the thorough/convincing level of the current experiments presented. - Only {ResNet-110, VGG-19} and CIFAR-10/100 are examined, albeit each with a large variety of lr/batch size. One would rather see more variety in model/data (like ImageNet models) - Where are the actual accuracy numbers achieved by those models? After changing the training protocol, are those results same competitive with SOTA numbers reported on the same model/dataset? Practically, a reduced generalization gap does not automatically grant a better testing set result. - All training techniques of SGD, such as momentum, are disabled. Also, both batch size and learning rate are constant without annealing. I wonder how those affect final achievable accuracy. - I believe the authors confused S_BS and S_LR definitions on lines 198-200.

Reviewer 2



The PAC bound in this paper is very similar to that in London 2017, which also derives an O(1/N^2) generalization bound. (London, B., "A PAC-Bayesian Analysis of Randomized Learning with Application to Stochastic Gradient Descent", NeurIPS 2017.) In the experiments, the authors seem to suggest that a larger learning rate is always better, using the pearson correlation coefficient to argue that there is a positive correlation between test accuracy and learning rate. Obviously, this reasoning breaks down at some point --- at some point using a large learning rate keeps you from learning --- and it is strange that the authors did not experiment with large enough learning rates to see this effect. Specific comments: - Equations in 5 are only true if we haven't seen training data batch n. If we are cycling through the training data, then these do not hold. - The author's assume that the stochastic gradients are Gaussian distributed. I don't know if this is a good assumption --- I can certainly think of simple contrived examples where the gradient is either +d or -d for some large value d. - The authors should cite two other important papers in this area, who do a much more detailed analysis of this subject: Smith and Le, "A Bayesian Perspective on Generalization and Stochastic Gradient Descent" 2018, and Smith and Le, "Don't Decay the Learning Rate, Increase the Batch Size", 2018. - The experiments show a relationship between the batchsize, learning rate, and test accuracy, but that in itself is not very surprising. Computing Pearson correlations does not seem appropriate because the relationship is not linear, and the p-values are overkill.

Reviewer 3



This paper presents a novel strategy for training deep neural networks with SGD, in order to achieve a good generalization ability. In particular, the authors suggest controlling the ratio of batch size to learning rate not too large. Extensive theoretical and empirical evidences are provided in the paper. Pros. 1. In general this paper is very well written and clearly organized. This paper brings new insights on training deep models to the community. The strategy has been well justified using both theoretical analysis and empirical evaluations. 2. Theoretical analysis is provided. In particular, a generalization bound for SGD is derived, which has a positive correlation with the ratio of batch size to learning rate. Tightness of the bound is also discussed. 3. A large number of popular deep models are trained and analyzed in the experiments. The Pearson correlation coefficient and the corresponding p values clearly support the proposed strategy. Cons. 1. Existing generalization bounds for SGD are discussed in Section 5. It will be helpful if the authors can discuss if the existing bounds involve learning rate and batch size or not. 2. A few typos should be corrected in final version. --Page 2: in terms of --Theorem 2: hold --Section 4.1: the elements in S_BS and S_LR ------------------------------------------------------------------------------------------- The authors have provided detailed and reasonable responses in their rebuttal. I still believe that this paper presents important theoretical results and brings new insights to the community. Thus, I vote for acceptance.

[Author Response · NeurIPS 2019]

The authors sincerely thank all the reviewers for their enormous effort and constructive comments. We will thoroughly review all related works, avoid typos, and carefully address all concerns in the final version.

**To Reviewer #1:**

**Q1:** *The work "Don't Decay the Learning Rate, Increase the Batch Size" suggested opposite theories.*

**A1:** We respectfully argue that the mentioned paper exactly supports our theory. This paper empirically demonstrates that when the rate of batch size to learning rate remains the same, the generalization is invariant, which is theoretically validated by our work.

**Q2:** *More varieties in model/data are needed.*

**A2:** We conducted more experiments of MLP on MNIST, AlexNet on CIFAR-10/100, and ResNet on ImageNet, during the rebuttal stage. All obtained results are consistent with the proposed theory and will be added to the final version.

**Q3:** *Actual accuracy after changing protocols. A reduced generalization gap does not grant a better test result.*

**A3:** We respectfully argue that the reported results are based on the test performance. Also, the highest accuracy is the actual accuracy of the models after changing protocols.

**Q4:** *The effect of disabling momentum and constant batch size and learning rate.*

**A4:** The effects of momentum and adaptive batch size and learning rate have been sufficiently studied (although only from the empirical view). By contrast, our paper studies the relationship between the generalization ability and the rate of learning rate to batch size from both theoretical and empirical aspects. It will be interesting to develop theories for momentum and adaptive batch size and learning rate. However, this is beyond the scope of this paper.

**To Reviewer #2:**

**Q5:** *The assumptions made to obtain this bound are not convincing.*

**A5:** Justifications for the assumptions are given below: (1) the gradients on one single data point, a mini-batch, and the whole training set are assumed to be unbiased estimations of the expected gradient, because all data points are independently drawn from the same distribution; (2) applying the large number theorem, it is rigorously correct that the gradient of one single data point is Gaussian distributed when the training set size is sufficiently large; and (3) existing experiments have demonstrated that the loss surface around the local minima is two-order smooth (cf. Appendix A.1).

**Q6:** *The relationship between generalization and the rate seems to be more complex than the authors suggest.*

**A6:** There are two terms in our bound which have positive and negative correlations with the rate of batch size to learning rate, respectively. However, we can prove that the positive correlation is dominant, when the parameter size $d$ is sufficiently large ($d > (|S|/\eta)/2\text{Tr}(A)$). The condition has been empirically validated in [22, 25, 26] for deep neural networks.

**Q7:** *The Hessian matrix changes drastically during learning, and it's not obvious how this fact changes the conclusions.*

**A7:** We respectfully argue that the Hessian matrix $A$ of the loss surface around global minima is invariant during learning. It only relies on the neural network architecture and the data distribution.

**Q8:** *The experimental result is not very surprising.*

**A8:** We acknowledge that some existing results suggested our finding, which are however only from the empirical aspect. Theoretically, the understanding is still elusive. Our contributions validate the intuitions theoretically and empirically. Moreover, our empirical studies are more comprehensive.

**Q9:** *PCC is only for linear models and the $p$-values are overkill.*

**A9:** We agree that PCC is designed for linear models, while it can still measure correlations. We further perform Spearman correlation test on the collected data, which is not based on any model. The results are in full agreements with our theory and will be added to the final version. Further, we respectfully argue that $p$-value is still important and even irreplaceable to measure the likelihood of an argument. The articles in *Nature* and *The American Statistician* concern the abuse of $p$-value that strictly uses $p = 0.05$ as the threshold of "statistically significant", especially when the sample size is small (e.g., smaller than 100). By contrast, the $p$-values in our paper are far below 0.05 and the sample size is sufficiently large. Specifically, the $p$-values for the rate are smaller than $10^{-88}$ and the sample size is $1,600$.

**To Reviewer #3:**

Thank you very much for your positive support to our work. We will thoroughly revise our paper to review all related works and to avoid typos.

[Meta-Review · NeurIPS 2019]

The paper proves a new upper bound for the generalization ability of algorithms trained by SGD, which demonstrate a negative correlation with the ratio of batch size to learning rate. The authors conducted experiments to verify the theoretical findings on a large number of models. The reviewers have mixed opinions on the paper. On one hand, the paper studies an important problem to the deep learning community, and the theoretical result has its uniqueness (e.g., regarding the ratio of batch size to learning rate), although some discussions on its correlation with previous PAC bounds are missing and some assumptions in the theory need more justifications. On the other hand, the suggestions resulting from the experiments (e.g., always increase the learning rate) seem not very reasonable and need more empirical verifications. Especially for more complex models for which local optima are not easy to find and are not very stable, the experimental observations could be totally different.